# Improving Model Generalization to Out-of-Domain Data in Histopathology: Leveraging Simple Techniques for External Validation

**Melanie Lubrano**[*1]                                              MELANIE.LUBRANO@OWKIN.COM
[1] *Owkin, Paris, France*

**Nathan Bigaud**[*1]                                              NATHAN.BIGAUD@OWKIN.COM
**Thalyssa Baiocco-Rodrigues**[*1]          THALYSSA.BAIOCCO-RODRIGUES@OWKIN.COM
**Fabien Brulport**[1]                                         FABIEN.BRULPORT@OWKIN.COM
**Alexandre Filiot**[1]                                      ALEXANDRE.FILIOT@OWKIN.COM
**David Lin**[1]                                                        DAVID.LIN@OWKIN.COM

## Abstract

This study explores strategies to enhance the generalizability of WSI classification models across diverse centers, crucial for clinical adoption. By leveraging optimal transport and Leave-One-Center-Out training policies, we achieve a 3% increase in the Youden Index, demonstrating their complementary roles in improving model performance. These approaches offer promising avenues for mitigating the need for extensive calibration samples, addressing challenges in external validation and ensuring robustness in clinical settings.

**Keywords:** WSI Classification, External Validation, Out-of-Domain, Calibration

## 1. Introduction

Because of the diverse processing steps involved in its generation, digital histopathology data exhibits significant heterogeneity across various centers. These steps range from sample preparation procedures such as staining and slicing, to the specifics of digitization like resolution and scanner type, resulting in notable shifts in data distribution. Consequently, ensuring the robust performance of AI models across new data distributions has become a standard practice in the field. This is typically achieved through testing trained models on what are termed "external cohorts" — datasets sourced from independent centers distinct from those used for training. External validation is particularly important when developing AI tools that will be used in clinical routine. Indeed, such tools should guarantee a minimum level of performance, regardless of any observed distributional shifts among different centers. Without these "guarantees", adoption of such tools in routine is impossible as it can significantly influence patient care

Ideally, achieving robust and generalizable models would be achieved using large and heterogeneous training datasets, reflective of the various potential data distributions. But digital pathology data are scarce, difficult to obtain, and often coming from one or few centers, limiting sources of variability.

---

[*] Contributed equally

MSIntuit (Saillard et al., 2023), a clinically validated AI algorithm designed for predicting Microsatellite Instability binary status (MSI or MSS) from H&E slides, proposes to ensure minimal performance standards across external centers through calibration. This involves utilizing a set of 30 MSI slides from each external center to calibrate model outputs and adjust classification thresholds accordingly. Nonetheless, calibration presents certain limitations. Firstly, gathering the necessary slides from each external cohort may pose challenges, especially for diseases with low prevalence. Secondly, it remains dependent on the selection of slides used for calibration, which may not fully capture the broadness of data distribution.

In this study, we propose an exploration of two strategies to enhance model generalizability on external data, thereby mitigating the necessity for extensive sets of calibration slides. Each proposition detailed subsequently, addresses distinct stages of the model and as such, are complementary.

## 2. Methods

### 2.1. Data

We used public and proprietary data to train and test our model. Each slide was associated with a binary label (MSI or MSS). A total of 2,200 H&E slides from various centers were used in the training set (18% MSI), and a total of 1,198 slides from 5 external cohorts were used as testing sets (34% MSI). Each slide was tiled in patches of 224x224 pixels at a resolution of 0.5 mpp. Features from each tile were extracted with a custom version of Phikon feature extractor (Filiot et al., 2023), trained on colorectal cancer slides.

### 2.2. Model and Training

As in (Saillard et al., 2023), Chowder Multiple Instance Learning was used with binary cross entropy. The training dataset was split into 15 stratified splits and cross validation was performed on each split. All the resulting trained models were ensembled to obtain prediction on the external test cohorts.

### 2.3. Evaluation and Baseline

As baseline, we measure the classification performances on each external cohorts using a standard calibration technique: 10 MSS and 5 MSI slides were randomly sampled from each external cohort and used only for calibration on their respective cohort, classification threshold was determined to reach a minimum of 0.9 sensitivity on the calibration data, reflecting the actual clinical use case (limiting false negatives).

### 2.4. Leverage 1: Use Optimal Transport to improve threshold calibration.

Optimal Transport (OT) (Peyré et al., 2019) defines the optimization problem which aims at minimizing the cost of transportation of a source distribution to match a target one. We used OT to map classification threshold computed on the training set (defined to reach Sensitivity>0.9) to calibration sets. To do so, the training set was bootstrapped into N subsets (N=1000) sampled to ensure a MSI prevalence of 30%. OT was used to map the

Table 1: Classification performances averaged on 5 external cohorts. 1000 bootstraps on each cohort and averaged.

| Leverage 1 Calibration | Leverage 2 Training Protocol | AUC [95%CI] | Youden Score [95%CI] | Sensitivity [95%CI] | Specificity [95%CI] |
|---|---|---|---|---|---|
| baseline optimal transport | random splits random splits | 0,929 [0.899-0.971] | 0,705 [0.646-0.775] 0,712 [0.64-0.808] | 0,867 [0,858-0.879] 0,889 [0.864-0.944] | 0,837 [0,768- 0,907] 0,822 [0,762-0.898] |
| baseline optimal transport | **LOCO** **LOCO** | 0,926 [0.903-0.965] | 0,711 [0.643-0.803]] **0,734 [0.658-0.815]** | 0,884 [0.836-0.912]] **0,911 [0.874-0,934]** | 0,827 [0,733-0,901] **0,823 [0,725- 0,903]** |

N thresholds to the calibration set of each external cohorts respectively. The N mapped thresholds were then averaged, resulting in final mapped threshold for each cohort to be used for classification of the test data.

### 2.5. Leverage 2: Apply a "Leave One Center Out" (LOCO) training policy

The cross validation splits are no longer generated randomly but sampled in a "leave one center out policy". The slides from one center are used either in training or in validation exclusively. The resulting models are then ensembled to obtain predictions on the external test data. Loss monitoring and Early stopping is used to ensure that the model does not overfit on the "left out" split.

## 3. Results

To assess MSI/ MSS classification performances in clinical context, we report Sensitivity and Specificity (see Table 1). The Youden Index (Sensitivity + Specificity -1) is used to capture the balance between the two. Using Optimal Transport alone or LOCO policy alone led to an increase of 0.7% and 0.6% of the Youden Index respectively while combining the two led to an increase of 3% showing their complementary role.

## 4. Discussion

In this paper we explored simple and efficient leverages to improve Out-of-Domain generalizability of AI models in the context of WSI classification. Each of the methods tackle separate portions of the classification pipeline, from the model training policy to the output calibration. These propositions can easily be customized for specific use cases: for instance, by using other covariates such as histological subtypes, biopsy/resection or tumor grade in the LOCO policy. Optimal transport has proven to be efficient to improve calibration, yet the ultimate goal is to have a "truly generalizable" model for which less calibration data or no calibration is needed. Further analysis will give more insight on the benefit of such methods, especially for indications with very low prevalence where calibration sets are extremely costly to acquire.

## Acknowledgments

We thank all the Owkin's Hackathon organizing team.

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
