# OpenReview forum: "Improving Model Generalization to Out-of-Domain Data in Histopathology: Leveraging Simple Techniques for External Validation"
_MIDL.io/2024/Short_Papers — MIDL 2024 Short Papers_

### Official Review · Reviewer_yZo8 · 2024-04-18

**Confidence:** 4
**Final Rating:** 3.5

**Review:**

The paper addresses using leveraging optimal transport and Leave-One-Center-Out training to improve generalization of a binary classification problem across histopathology domains.

Strengths:
-	Using existing well-known techniques
-	Performance reported with confidence intervals

Weaknesses:
-	Few details provided about the data, which public dataset was used? How are the domains different? It is difficult to say anything about out of domain generalization if no information about the domains is given
-	The paper is not self contained, methodologies like Phikon feature extractor and Chowder Multiple Instance Learning cannot be assumed to be known by the average reader
-	Unclear why specificity of optimal+LOCO is highlighted in bold, here the baseline+random has the best results
-	A bit surprising even for a short paper to have only 3 references

I think the work is relevant for the conference but many details are missing, I would prioritize other papers that are more self-contained therefore my conservative score.

---

### Decision · Program_Chairs · 2024-04-26

Accept